# CFE2: Counterfactual Editing for Search Result Explanation

## ABSTRACT

Search Result Explanation (SeRE) aims to improve search sessions' effectiveness and efficiency by helping users interpret documents' relevance. Existing works mostly focus on factual explanation, i.e. to find/generate supporting evidence about documents' relevance to search queries. However, research in cognitive sciences has shown that human explanations are contrastive i.e. people explain an observed event using some counterfactual events; such explanations reduce cognitive load and provide actionable insights. Though already proven effective in machine learning and NLP communities, there lacks a strict formulation on how counterfactual explanations should be defined and structured, in the context of web search. In this paper, we first discuss the possible formulation of counterfactual explanations in the IR context. Next, we formulate a suite of desiderata for counterfactual explanation in SeRE task and corresponding automatic metrics. With this desiderata, we propose a method named **C**ounter**F**actual **E**diting for Search Research **E**xplanation (**CFE2**). CFE2 provides pairwise counterfactual explanations for document pairs within a search engine result page. Our experiments on five public search datasets demonstrate that CFE2 can significantly outperform baselines in both automatic metrics and human evaluations.

## KEYWORDS

Information Retrieval, Counterfactual Explanation

**ACM Reference Format:**
Anonymous Author(s). 2018. CFE2: Counterfactual Editing for Search Result Explanation. In *Proceedings of Make sure to enter the correct conference title from your rights confirmation emai (Conference acronym 'XX)*. ACM, New York, NY, USA, 11 pages. https://doi.org/XXXXXXX.XXXXXXX

## 1 INTRODUCTION

Search systems such as those used within search engines and product search have played a central role in the way people acquire information. Albeit effective, recent advancements in neural retrieval methods [18] provides little human-interpretable evidence of the underlying reasoning they use to determine relevance [41, 42]. Providing explanations on Search Engine Result Page (SERP) to explain documents' relevance has been shown to improve search sessions' efficiency and users' trust towards the system [25, 33].

In line with the general work of explainable AI literature [26, 51], these explanations are factual in nature. The goal of these factual explanations is to find/generate supporting evidence for model's decisions, i.e. to answer the "Why $P$" question. Most existing works currently utilize factual explanation for the task of Search Result Explanation (SeRE), i.e. to find/generate supporting evidence to explain documents' relevance w.r.t. the search query. Examples of factual explanations are snippet-based [4, 49] methods and NLG-based methods [32].

However, research in cognitive science has shown that human explanations are contrastive [26], i.e. people explain an observed event using some counterfactual events/contrast cases.[1] Instead of answering the "Why $P$" question, counterfactual explanation answers the "Why $P$ rather than $Q$" question. For example, Alice gets rejected in credit card application and the counterfactual explanation is, were her credit history to be one year longer, the application would have been approved. Such explanations prune the space of causal factors to reduce the users' cognitive load [13, 19] and provide actionable suggestions [51]. Although counterfactual explanations have proven effective in other communities [36, 51], there lacks a well-established formulation of counterfactual explanation for search result explanation task. Moreover, it is unclear what is the desiderata of such explanations and corresponding evaluation metrics properly evaluate such explanations.

Seeing this gap, we aim to investigate the impact of counterfactual explanations for the task of SeRE. Within the scope of this work, we discuss the following research questions:

**RQ1. Possible formulations of counterfactual explanation problem in the context of web search.**

In Section 3, we motivate the problem by reviewing counterfactual explainability literature from psychology and cognitive science communities, and discuss the possible formulation of counterfactual explanations in the web search setting. We propose the problem formulation of using counterfactual queries as explanations, as such explanations are directly related to the user activity, thus enabling users to interact with and to interpret the system via potential query reformulation in the rest of the search session.

With this formulation, we study pairwise counterfactual explanation, where counterfactual query/explanation is provided as explanation to the (initial query, top-ranked document, lower-ranked counterfactual document) triplet, i.e. were the initial query to be changed to the counterfactual query, the counterfactual document would be ranked higher than the initially higher-ranked document. Providing such explanation not only explains the initial pairwise relevance relation, but also enables actionability where users can interpret the system, and potentially refine the search result with the counterfactual query.

**RQ2. Design Principles and Evaluation for Counterfactual Explanations**: What are the desired properties of counterfactual explanations for the task of SeRE? How do we evaluate the quality of machine-generated counterfactuals?

In Section 4 we formulate the desiderata for counterfactual explanations in the context of web search. With this desiderata, we propose a suite of automatic evaluation metrics to comprehensively evaluate the quality of counterfactual explanations in the IR context.

---

[1] Although the definitions of 'Counterfactual' and 'Contrastive' have minor difference in literature [26, 36, 53], they can be seen as equivalent in our problem formulation of using counterfactual query as explanation, as a counterfactual query will naturally lead to a different search result, i.e. a contrast case.

*Conference acronym 'XX, June 03–05, 2018, Woodstock, NY*
2018. ACM ISBN 978-1-4503-XXXX-X/18/06...$15.00
https://doi.org/XXXXXXX.XXXXXXX

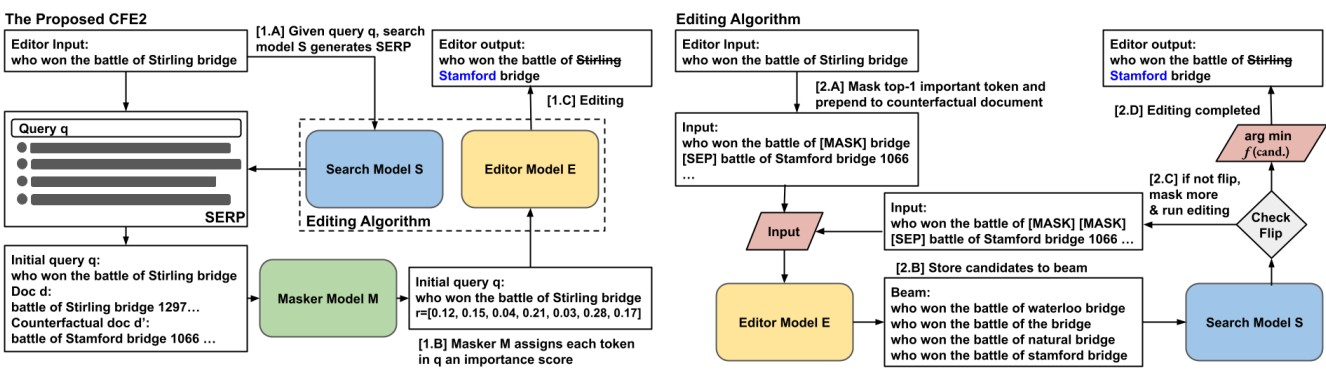

(a) A sample workflow of one edit.

(b) A closer look at the editing algorithm.

**Figure 1: An overview of CFE2 . Fig 1a shows a sample workflow of one edit. CFE2 takes a query as input, then [1.A] search model S generates a SERP; for a document pair $(d, d')$, [1.B] masker M generates importance score, then [1.C] editor E performs editing. Fig 1b shows a more detailed editing loop. [2.A] the top-1 important token from query is masked then the masked query is prepended to counterfactual document and input to editor E; [2.B] the editor predicts word and stores the candidates to beam $\mathcal{B}$; then checks flip. If not flip, [2.C] it will mask one more token and run word prediction/decoding again; if flip, then [2.D] editing is complete and editor will output the counterfactual query with lowest perplexity, serving as a counterfactual explanation to initial $(q, d, d')$ triplet.**

**RQ3. Method for Counterfactual Explanations**: How do we design an algorithm to provide pairwise counterfactual explanations for document pairs within the current SERP?

In Section 5 we propose a model named **C**ounter**F**actual **E**diting for Search Result **E**xplanation (**CFE2** ), overviewed in Figure 1. Given (initial query, top-ranked document, lower-ranked counterfactual document), CFE2 edits the initial query into a new query such that the previously lower-ranked document is now ranked higher by the system. This edited counterfactual query serves as a counterfactual explanation to the documents' pairwise relevance relation w.r.t. the initial query. CFE2 outperforms baselines in automatic metrics and human evaluations on public IR datasets (Section 7).

The proposed CFE2 has additional strengths: (1) CFE2 works in a black-box/model-agnostic setting, and therefore does not depend on specific retrieval model choices. (2) CFE2 can provide counterfactuals that are of minimal modifications compared to the initial query $q$, meeting the definition of counterfactual explainability. (3) CFE2 uses off-the-shelf transformer-based word prediction model as backbone but is also easily extensible by finetuning on the target corpus with minimum extra computational effort. (4) CFE2 is lightweight, meeting the low-latency requirement of SeRE task.

## 2 RELATED WORK

We mainly introduce two lines of related works and leave a longer discussion to the final version.

**Explainable AI and search.** Existing explainable AI methods can be broadly divided into two categories: model-intrinsic methods, where the decision model is interpretable by design; and model-agnostic methods where explanations are generated from a specific explanation model that is different from the decision model [1, 22]. Model-intrinsic explainable systems often introduce extra model complexity and challenge the search/retrieval systems' low latency

**Table 1: Comparison with existing SeRE methods, ✓/✗ means depending on specific model configurations.**

| Methods | Properties | | | |
|---|---|---|---|---|
| | Model-agnostic | Counter-factual | Minimum Modifi. | Comp. Complex. |
| Perturbation-based methods [41, 42, 50] | ✓ | ✗ | ✗ | Medium |
| Extractive Snippets [4, 49] | ✓/✗ | ✗ | ✗ | Low |
| Abstractive Snippets [6] | ✓ | ✗ | ✗ | High |
| Query-biased Summary [46, 61, 62] | ✓ | ✓/✗ | ✗ | High |
| Query-biased Generation [32] | ✓ | ✓/✗ | ✗ | High |
| CFE2 (Ours) | ✓ | ✓ | ✓ | Medium |

design principle [21, 22, 47]. In this work, we opt to study model-agnostic explanation for search/retrieval, where we assume no knowledge w.r.t. architecture and/or parameters of the underlying blackbox retrieval model. In this work, the proposed explanation method CFE2 does not utilize the knowledge from the underlying retrieval model, i.e. architecture and/or parameters, therefore falls into the category of model-agnostic explanations; and the explanation methods that utilize the decision model's knowledge, e.g. gradient attribution methods, are not directly comparable.

Existing explainable search works mainly focus on answering questions such as (1) why is this document relevant to the query (pointwise) i.e. to explain document's relevance, (2) why is this document ranked higher than the other (pairwise)? i.e. to explain the ranking. or (3) why is this set of documents returned (listwise)? Works on pointwise explanation broadly falls into three categories: (a) *Snippet-based methods* provide snippets (sentences or sentence

fragments) as explanations [4, 46, 49, 62] and have been provided in mainstream search engine applications (Google, Bing, etc.) [52]. (b) *NLG methods* propose to generate natural language explanations separately to explain documents' relevance. GenEx [32] utilizes an encoder-decoder structure to generate terse explanations in addition to extractive snippets. (c) *Perturbation-based methods* perturb the input (query and document) to create saliency maps to explain documents' relevance. Verma and Ganguly [50] and Singh and Anand [41] adapt LIME [34] to generate word-based explanations to explain one single document's relevance w.r.t. the search query. Very few works are centered on pairwise explanations. Singh and Anand [41] propose EXS to explain pairwise ranking preference. To the best of our knowledge, LiEGe [63] is the only work that aims to jointly explain the entire list of retrieved documents.

In this work, we also aim to explain a neural ranker's pairwise preference. Instead of generating/editing counterfactual documents like prior works [5, 36, 41], we focus on providing counterfactual explanations from the query perspective, i.e. *Document $d'$ would be ranked higher than $d$ if you modify your query $q$ to $q'$*. Thus, our method can provide user with ways to understand model's reasoning and take actions such as reformulating query to improve search session's effectiveness. As suggested by previous works in web search [2, 14, 27, 37], it is more relevant to suggest alternate queries that are informative, actionable and can assist users in satisfying their information needs.

**Factual and counterfactual explanation.** Existing factual explanation [43, 51] methods mostly aim to find/generate supporting evidence to explain model predictions. All works discussed were examples of factual explanations. Some representative methods include input feature attribution methods [40, 55] and natural language explanations [58]. From the causal inference perspective, counterfactual explanation doesn't directly answer the "why" part of prediction [51], but instead answers the what-if question, i.e. *what would the model predict if the current input is changed*? As discussed by Miller [26] and Lipton [19], human explanations are contrastive from a cognitive science perspective; people explain an observed event using some counterfactual events. Multiple works from NLP community [13, 36, 60] have discussed the formulation and usage of counterfactual explanations.

**Contrastive editing.** Ross et al. [36] proposed Minimum Contrastive Editing (MiCE) method to explain NLP models. They prepend classification labels to input text and apply a binary search method to find minimum modification to input text to alter the prediction of the NLP classifier. Our method shares similar design choices but is different from the following perspective: (a) our method is designed for ranking task compared to the classification task; (b) MiCE operates on longer text input and uses a span infilling approach for editing while our method operates on shorter queries and uses a different token prediction approach. (c) MiCE uses gradient attribution method to select important tokens to mask, which requires full knowledge w.r.t. the prediction model. In contrast, our method utilizes a separate Masker model **M** to select important tokens thus operates in a blackbox setting and can serve as a plug-in explanation method to existing search/retrieval models.

## 3 MOTIVATION AND PROBLEM STATEMENT

**Motivation of counterfactual explanation.** We draw inspiration from prior works on counterfactual explanations in the psychology and social science literature [11, 19, 26]. Hilton [11] poses an important insight that one does not explain events per se, but that one explains why the puzzling event occurred in the target cases but not in some counterfactual contrast cases. Denote $P$ as fact, and $Q$ as the counterfactual case, Lipton [19] and Miller [26] argue that the explainer does not need to consider all causes of an event, but should focus on those causes that are relevant to the counterfactual case. For instance, instead of explaining "Why $P$?", it is more effective to explain "Why $P$ rather than $Q$?". This approach prunes the space of causal factors to reduce the user's cognitive load. Additionally, it provides useful actionable suggestions, allowing the user to better interpret the system by further interacting with the system to refine the predictions. In the context of web search, counterfactual explanations are useful as the user can easily interact with the system by reformulating queries and refining the search results, eventually improving the effectiveness and efficiency of their information-seeking sessions [24, 39, 56].

**Formulation of counterfactual explanation in web search.** To the best of our knowledge, there has not been a strict formulation for defining and structuring counterfactual explanations in the context of web search. We consider a vanilla search system setting, where the user issues query $q$, and the system returns a list of individual document $d \in \mathcal{D}$, $\mathcal{D}$ denotes the document collection. We denote search system's input as $(q, d)$ pair, and the system predicts a relevance score rel.$(q, d)$. The counterfactual event can be of two formulations:

(1) The user would have issued a different query, i.e. *counterfactual query*, and subsequently, the system would have retrieved a different list of documents, i.e. *counterfactual documents*.
(2) A currently top-ranked document, would have been ranked to a lower position by the system, if the document were to be changed in particular ways, i.e. a *counterfactual document*.

In brief, counterfactuality in Formulation (1) is from the *counterfactual query*, while in Formulation (2) it is from the *counterfactual document*. In machine learning systems, to be tangible to end users, the provided explanations should relate to the user's own activity, and be scrutable, actionable, and concise [3, 9, 10, 48]. With this setup, the user can leverage the explanation to prune the action space, and further interpret/interact with the complex system. Since a search engine user can issue alternative queries but cannot change the document(s) in the collection, we propose that Formulation (1) is more suitable for web search setting. With this formulation, the user can leverage the counterfactual explanation to interpret the black-box web search system, to reformulate queries to fulfill their information needs [7, 39], potentially improving the effectiveness and efficiency of the search session [24, 56].

**Pairwise counterfactual explanation.** Given an initial query and a list of documents, we can further provide counterfactual explanations for:

(1) The whole list of documents.
(2) A pairwise relation between the initially top-ranked document and one counterfactual document that was initially at a lower

**Table 2: Table of Notations**

| | |
|---|---|
| $q, d, \mathcal{D}$ | query, document, set of all documents |
| $q', d'$ | counterfactual query, counterfactual document |
| **S, M, E** | Search Model, Masker Model, Editor Model |
| rel.$(q, d)$ | relevance score of $(q, d)$ with the search model **S** |
| $\pi(q)$ | List of documents for query $q$, sorted by rel.$(q, d)$ |
| $f(q, d, d') \rightarrow q'$ | explainer model to produce counterfactual query $q'$ |
| $\mathcal{B}, b$ | Beam, beam size |
| ppl.$(\cdot)$ | Perplexity Calculator, computed by the mean perplexity of a sequence; a smaller value indicates the sequence is "fluent" from natural language perspective |

rank position, but would be ranked at a higher position were the counterfactual query to be issued.

These two settings align with the listwise and the pairwise explanation paradigm of explainable search [1]. Since a minor perturbation to the initial query may result in a completely different list of documents being retrieved, which is intractable as document collection size increases, we exclusively study pairwise counterfactual explanations in the scope of this work. Specifically, we study the pairwise relevance relations within the same current search engine result page, e.g. one document at an initially higher rank position (more relevant), i.e. original document, and one document at a lower rank position (less relevant), i.e. counterfactual document.

**Problem definition of counterfactual explanation for web search.** We show a summary of notations in Table 2. Denote rel.$(\cdot, \cdot)$ as a function determined by search model **S** to assign a relevance score for each $(q, d)$ pair, and $f(q, d, d')$ as the explainer model. We formally define our problem, as follows:

> **Problem Definition**: Given the initial query $q$ and document pair $(d, d')$ in a SERP, and rel.$(q, d) >$ rel.$(q, d')$, the goal is to design explainer model $f(q, d, d') \rightarrow q'$, such that rel.$(q', d') >$ rel.$(q', d)$.

This query $q'$ is counterfactual to the initial query $q$, and serves as a counterfactual explanation to the initial triplet $(q, d, d')$. [2] This counterfactual explanation provides explainability of why $d$ is ranked higher than $d'$ by the system. In addition, it also providing *actionability* (e.g. query reformulation) where the user can leverage the counterfactual query $q'$ to refine the ranklist $\pi(q)$, or to retrieve more documents similar to $d'$.

## 4 EVALUATION PRINCIPLES AND METRICS

Previously in Section 3, we have discussed the possible formulations of counterfactual cases in the context of web search, and formalized the definition of counterfactual explanation for web search. However, there has not been a well-structured framework to evaluate counterfactual explanations in this problem setting. In this section, we first discuss desiderata for counterfactual explanations (Section 4.1), then design corresponding automatic evaluation metrics in Section 4.2.

---

[2]For the rest of the paper, we use counterfactual query and counterfactual explanation interchangeably.

### 4.1 Desiderata for Counterfactual Explanation for Web Search

Prior studies [12, 19, 26, 51] have established a comprehensive list of desiderata for explainable systems. In this work, we account for these important factors, and adapt them to the specific context of counterfactual explanations for search systems. An ideal counterfactual explanation method should meet the following desiderata:

- **Max #Flips**: The counterfactual explanation method should be effective in achieving its objective[3], i.e. it ensures that generated counterfactual query could flip the ranking order of the initially higher-ranked document with the counterfactual document. This effectiveness should hold true for all or most of the random document pairs in the collection.
- **Closeness**: An ideal counterfactual explanation method should generate counterfactual queries that are semantically close to the initial query. This closeness can make the explanations more *intelligible* and more *actionable* for end users.
- **Fluency**: Web search queries are often short and concise. The counterfactual query and hence the explanation should have similar fluency to the initial query. Over-complex or poorly phrased explanations are deemed less useful to users [26]. On the other hand, over-simplified counterfactual queries may be incoherent, grammatically incorrect, and may fail the purpose to serve as meaningful explanations [36, 57].
- **Low Latency**: The counterfactual query generation system should provide explanations in real-time [21, 47]. This ensures the counterfactual queries are *interactive* and *actionable* for users.

### 4.2 Automatic Evaluation Metrics

We operationalize the desiderata into automatic metrics to evaluate the effectiveness of any proposed or existing methods:

- **FlipRate**: the proportion of the edited counterfactual queries that satisfy the objective i.e. rel.$(q', d') >$ rel.$(q', d)$. This metric captures the factor Max #Flip.
- **CosSim**: Cosine Similarity computed between the vector representation of $q$ and $q'$ used by search model **S**. It is bound to $[0, 1]$ where higher CosSim indicates higher similarity (Closeness).
- **BERTScore-F1**: BERTScore [66] evaluates the semantic similarity between two sequences using contextualized representations from a transformer-based model. It is bound to $[0, 1]$ where a higher value indicates higher semantic similarity (optimizes for Closeness); we use BERTScore-F1 to balance precision and recall.
- **RelFluency**: we first use a pre-trained language model GPT-2 [31] to compute the mean perplexity of the generated counterfactual and the initial query, respectively; denote $q_{(i)}$ as the $i$-th token in query $q$, and $P_{\text{LM}}(q_{(i)})$ as the language model's probability of predicting token $q_{(i)}$ given the context of $(i-1)$ tokens:

$$\text{ppl.}(q) = \left( \sum_{i \in |q|} -P_{\text{LM}}(q_{(i)}) \log P_{\text{LM}}(q_{(i)}) \right) / |q| \quad (1a)$$

$$\text{RelFluency} = \text{ppl.}(q') / \text{ppl.}(q) \quad (1b)$$

Here ppl.$(\cdot)$ denotes the mean perplexity of a text sequence. RelFluency=1.0 is an ideal case indicating the counterfactual

---

[3]also referred to as *soundness* [19, 26]

query is of the same fluency as the initial query [36]; while an extreme RelFluency score means the counterfactual query deviates from the initial query.[4]

- **Runtime**: We also measure the average wallclock time per edit (captures Latency).

Additionally, later results from human evaluation (Section 7.2) indicate these metrics are coherent with human understanding.

## 5 THE PROPOSED CFE2

Based on desiderata in Section 4.1, we propose an approach to generating counterfactual queries as explanations, named **C**ounter**F**actual **E**diting for Search Result **E**xplanation (**CFE2** ). CFE2 operates by iteratively editing the initial query by dropping or replacing tokens to obtain the counterfactual query. It comprises three components: Search Model **S**, Masker Model **M** and the Editor Model **E**. A sample editing workflow is as follows:

(1) The Search Model **S** returns a SERP where $d$ is top-1 ranked document.
(2) The Masker **M** assigns each token in $q$ an importance score.
(3) For a counterfactual document $d' \in (\pi_q \setminus d)$ (other documents returned by SERP), Editor **E** will edit the initial query $q$ to generate $q'$.

An overview of our framework and a sample editing workflow is presented in Figure 1. A detailed algorithm is in Algorithm 1.

### 5.1 Search Model S

A key advantage of our counterfactual explanation framework is its model-agnostic nature. It operates independently of the underlying search algorithm, allowing for flexibility in choosing any model. In this paper, we conduct all our experiments using CL-DRD [64], a state-of-the-art single vector dense retrieval model. For query $q$, **S** assigns each document in $\mathcal{D}$ a relevance score and constructs a ranklist $\pi(q)$ accordingly. The SERP for which we provide explanations consists of both ranklist $\pi(q)$ and query $q$.

### 5.2 Masker Model M

Masker model **M** needs to identify important tokens in $q$. Then these tokens can be replaced with tokens that alter the meaning of the query, such that the counterfactual document can be ranked higher over the initially higher-ranked document. To do this, we leverage a method inspired by ColBERT [15], a dense retrieval model originally designed for neural ad-hoc retrieval.

Given a query $q = q_{(1)}q_{(2)} \ldots q_{(l)}$ of $l$ tokens and a document $d = d_{(1)}d_{(2)} \ldots d_{(n)}$ of $n$ tokens, we can get a bag of contextualized token embeddings for each token in $q$ and $d$ from output of BERT [8], this we denote by **v**. For each token $q_{(i)}$ in $q$, we estimate its importance score $r_i$ by computing as follows:

$$r_i = \max_{j \in |n|} \mathbf{v}_{q_{(i)}} \cdot \mathbf{v}_{d_{(j)}}^{\mathbf{T}} \tag{2}$$

where $\mathbf{v}_{q_{(i)}}$ and $\mathbf{v}_{d_{(j)}}$ denote the contextualized token embedding for $i$-th token in $q$ and $j$-th token in $d$, respectively; max denotes a max pooling operation over all tokens in $d$. Here the pooling

[4]We should note that perplexity metric ppl. $(q)$ in Equation (1a) tends to get lower values for longer sequences because of the denominator $|q|$, this is also reflected in our experimental results in Table 4, RelFluency<0 may be due to the methods tend to provide long and tedious counterfactual explanations.

---

**Algorithm 1** Detailed Editor Algorithm

---

**Input:** Initial query $q = q_{(1)}, q_{(2)}, \ldots, q_{(l-1)}, q_{(l)}$, $\mathbf{M}(q, d) = r_1, r_2, \ldots, r_l$, initially top-1 ranked document $d$, desired relevant document $d'$, rel.$(q, d)$>rel.$(q, d')$ and Perplexity Calculator ppl.$(\cdot)$,

**Output:** Edited Query $q'$

1: Initialize an empty beam $\mathcal{B}$
2: **while** $i$ in range(1, $n$+1) **do**
3:      Mask top-$i$ important tokens in $q$ to get $q^{\mathbf{M}}$
4:      Prepend masked $q^{\mathbf{M}}$ to desired document $d'$ and input to **E**
5:      Predict the top-$b$ tokens for timestep 1 and save $q^{\mathbf{E}_1}$ and $Prob(q^{\mathbf{E}_1})$ to beam $\mathcal{B}$
6:      **for** timestep $t$ in range(2, $i$+1) **do**
7:          Compute $Prob(q^{\mathbf{E}_t})$ by Eq. 3 and Eq. 4.
8:          Sort $Prob(q^{\mathbf{E}_t})$, save top-$b$ $q^{\mathbf{E}_t}$ and $Prob(q^{\mathbf{E}_t})$ to $\mathcal{B}$.
9:      **end for**
10:      $\Omega \leftarrow q_c \in \mathcal{B}$ that satisfy rel.$(q', d')$>rel.$(q', d)$
11:      **if** $\Omega \neq \emptyset$ **then**
12:          $q' = \arg \min_{q_c \in B} \mathrm{ppl.}(q_c)$.
13:          break
14:      **else if** $\Omega = \emptyset$ **then**
15:          $i = i + 1$
16:      **end if**
17: **end while**
18: return Null if not $q'$ else $q'$

---

operation catches the most relevant document token to the query token $q_i$; and different $r_i$ indicates the contribution of each query token to the overall ranking score of $(q, d)$ pair, as computed by rel.$(q, d) = \sum_{i=1}^{l} r_i$. Here we use $r_i$ as an approximation of token level importance of the query $q$.

### 5.3 Editing Algorithm

Given the input triplet $(q, d, d')$, the task of counterfactual query editing is to edit initial query $q$ to a counterfactual query $q'$. We formulate counterfactual query editing task as a generation task for a predefined number of $n$ tokens. Specifically, we try to find a counterfactual query $q' = q'_{(1)}, q'_{(2)}, \ldots q'_{(l)}$ from initial query $q = q_{(1)}, q_{(2)}, \ldots q_{(l)}$. First, we mask a few important tokens in $q$ as measured by $r_i$, which is computed by the masker model. Next, we run word prediction algorithm (also known as Masked Language Modeling task) to predict the masked tokens [MASK] in an autoregressive manner. Since we may have multiple [MASK]s to predict and the candidate space is the whole vocabulary $\mathcal{V}$, it is infeasible to track all possible edits and rate them. Therefore, we leverage Beam Search to prune the search space. In addition, given that we want to ensure Closeness (Section 4.1), we formulate the algorithm as an iterative process, i.e. we start from only one [MASK] token, and gradually increase the number of [MASK]s until we find an edit that achieves the objective of returning document $d'$ at a higher relevance score than the initially higher ranked document $d$. We describe the complete editing algorithm to Algorithm 1. Our method can be divided into three steps:

- **Step 1: Decoding** (line 3-10): We start with replacing top-$n$ important tokens as identified by **M** with [MASK] to get the

masked query $q^M$; Then, we prepend $q^M$ to the counterfactual document $d'$ and use the concatenated sequence as input to Editor model **E**. **E** will then run pre-specified transformer model to predict the masked tokens in an autoregressive manner.

To be more specific, at the start of timestep $t$, the beam $\mathcal{B}$ maintains a beam of $b$ possible edits $q^E_{t-1}$ from the previous timestep $t-1$. For each existing $q^E_{t-1}$, the probability of token $i$ being predicted at current timestep $t$ is computed by normalizing the logits over token $i$ with a softmax function

$$Prob(i|q^E_{t-1}) = \frac{\exp(\text{logit}_i)}{\sum_{j=1}^{|\mathcal{V}|} \exp(\text{logit}_j)} \tag{3}$$

where the denominator is computed over the whole vocabulary $\mathcal{V}$. Then for each existing possible edit $q^E_{t-1}$ in $\mathcal{B}$, we add one of the corresponding $b$ most probable tokens, resulting in $b \times b$ new possible edits. For these new possible edits, we compute the new probabilities

$$Prob(q^E_{t+1} = [q^E_t; i]) = Prob(q^E_t) \cdot Prob(i) \tag{4}$$

We sort the possible edits at the current timestep by the newly computed probabilities from Equation (4) and again select the top-$b$ candidate edits to refresh the beam $\mathcal{B}$. Then we move to the next timestep $t$+1.

- **Step 2: Checking** (line 11-13): After one round of editing is completed, we check all possible edits in $\mathcal{B}$. If there are edits that can flip the results, i.e. rel.$(q', d') >$ rel.$(q', d)$, we choose among these edits the one with the lowest perplexity score computed by the Perplexity Calculator ppl.$(\cdot)$.
- **Step 3: Iterative Search** (line 2-17): We start from masking one token and perform step 1-2. If there are no edits in $\mathcal{B}$ that can flip the results, we mask one more token and keep on repeating step 1-2 until an available edit is found or there are no more tokens from $q$ that can be masked.

Our editing algorithm's design reflects the desiderata: (1) we start the search for counterfactuals by modifying one token at a time to meet **Closeness**; (2) we stop the search when a counterfactual can flip the result such that the principle of **Max #Flips** is satisfied; (3) among the candidates that can flip the result, we choose the counterfactual query with lowest perplexity. This choice ensures fluent counterfactual queries. (4) We use word prediction model as the backbone for Editor **E**. In our implementation, we use a lightweight `DistilBERTForMaskedLM` (66M), which has a fully connected layer on top of a DistilBERT for word prediction. This choice meets the **Low Latency** principle. Notably, it can also be replaced with other word prediction algorithms (e.g. `BERTForMaskedLM`, etc.). In addition, it is known that large-scale transformer-based language models can benefit from further finetuning on target domain corpus [8, 35]. Thus we also finetune `DistilBERTForMaskedLM` on the target corpus. To distinguish from the one using off-the-shelf LM, we name our methods as CFE2 and CFE2+ , respectively.

## 6 EXPERIMENTS SETUP

**Datasets.** We use the well-established MS MARCO passage ranking dataset [29] together with three information retrieval datasets of different domains from BEIR collection [44]: NQ [16], FiQA [23], SciFact [54] and FEVER [45]. A summary of dataset statistics is

**Table 3: Summary of datasets**

| Dataset | Train #Pairs | Dev #Query | Test #Query | Test #Corpus | Avg. Lengths Query | Avg. Lengths Doc. |
|---|---|---|---|---|---|---|
| MS MARCO | 39.7m | - | 7.0k | 8.8m | 6.0 | 56.0 |
| FEVER | 140.1k | 6.7k | 6.7k | 5.4m | 8.1 | 78.9 |
| NQ | 132.8k | - | 3.5k | 2.6m | 9.2 | 78.9 |
| FiQA | 14.1k | 0.5k | 0.5k | 57.6k | 10.8 | 132.3 |
| SciFact | 0.9k | - | 0.3k | 5.2k | 12.4 | 213.6 |

referred to Table 3. For MS MARCO dataset, we use the official dev set as our test set; for FEVER and FiQA dataset, we combine queries in dev set and test set to use for test; and we use the NQ and SciFact test set from BEIR.

For each query in our testset, we first use search model **S** to retrieve top-5 passages from passage collections; then we construct the test triplets $(q, d, d')$ using top-1 passage as $d$ and the rest four passages as $d'$, leading to 20.4k, 53.3k, 4.6k, 1.8k and 1.2k test triplets for MS MARCO, FEVER, FiQA, NQ and SciFact respectively. CFE2 does not rely on relevance judgments for training so we do not make use of the train pairs for our method.

**Compared methods.** We conduct experiments with the following methods that are relatively closer to our problem setting:

- **MaxFlip**: we split the counterfactual document $d'$ into multiple sentences, and select the sentence with the lowest perplexity from the sentences that can flip the pairwise relevance from **S**.
- **Summary**: we use a Seq2Seq model finetuned on summarization datasets to summarize the counterfactual document as the counterfactual query.
- **GenEx** [32]: GenEx is originally designed to generate terse abstractive explanations given $(q, d)$ pair and is trained using $(q, d, e)$ triplets where $e$ is golden reference explanations. We train GenEx with only $(q, d)$ pairs from trainset as we do not have access to gold references $e$ from the dataset.
- **Mask-only**: this is an ablation of the proposed method without the Editing phase, i.e. we only replace the [MASK] token(s) from the masked query with [PAD] iteratively according to the token weights computed by ColBERT model, until the counterfactual query can flip the pairwise relevance from **S**.
- **CFE2** : proposed method with off-the-shelf word prediction model `DistilBERTForMaskedLM`.
- **CFE2+** : `DistilBERTForMaskedLM` is finetuned on the corpus of the target dataset. We construct the training corpus using the passages from target collections and train the model with standard masked language model objective.

**Implementation details.** We use `t5-base` finetuned on XSUM dataset [28] for `Summary` baseline, with 1e-4 learning rate. We replicate the structure of `GenEx` and disable the components that does not fit our task, and correspondingly train with the same hyperparameters reported by the original paper. For Editing w/ FT, i.e. CFE2 , we use a standard 0.15 mask ratio, 128 block size, 2e-5 learning rate and train 3 epochs on each dataset's corpus. In the decoding stage of CFE2 , we set beam size $b$ to 10 to balance performance and computational complexity. The choice of beam size's effect is studied in Section 7.3 and results reported in Figure 3. We use the official checkpoint of CL-DRD and ColBERT for the Search model **S** and Masker Model **M**. For evaluation, we use `gpt2-large` [31] to

**Table 4: Evaluation results of our method compared to baselines. Runtime is measured by #seconds/edit, we highlight the best performance among the models and † denotes statistically significant with paired t-test on all test pairs at 0.05 level compared to the best baseline. Significance test of Fluency is conducted upon |Fluency-1|.**

| Dataset | Metric | Baseline Methods | | | | Proposed Methods | |
|---|---|---|---|---|---|---|---|
| | | Mask-only | MaxFlip | Summary | GenEx | CFE2 | CFE2+ |
| MS MARCO | FlipRate | 0.832 | 0.986 | 0.972 | 0.931 | **0.998** | **0.998** |
| | CosSim | 0.885 | 0.855 | 0.889 | 0.902 | 0.909† | **0.910** |
| | BERTScore-F1 | 0.779 | 0.833 | 0.849 | 0.878 | **0.933**† | **0.933**† |
| | RelFluency | 4.245 | 0.245 | 0.304 | 0.385 | 1.774 | **1.563**† |
| | Runtime (sec/edit) | 0.210 | **0.119**† | 0.932 | 0.397 | 0.258 | 0.258 |
| NQ | FlipRate | 0.852 | 0.996 | 0.984 | 0.923 | **0.999** | 0.998 |
| | CosSim | 0.892 | 0.848 | 0.885 | 0.895 | 0.934† | **0.937**† |
| | BERTScore-F1 | 0.728 | 0.835 | 0.834 | 0.875 | **0.962** | 0.946† |
| | RelFluency | 3.236 | 0.420 | 0.952 | 1.554 | 1.159† | **1.012**† |
| | Runtime (sec/edit) | **0.242** | 0.247 | 1.285 | 0.441 | 0.257 | 0.257 |
| FiQA | FlipRate | 0.888 | 0.996 | 0.973 | 0.899 | **0.999** | 0.998 |
| | CosSim | 0.924 | 0.832 | 0.889 | 0.849 | 0.957† | **0.959**† |
| | BERTScore-F1 | 0.805 | 0.849 | 0.850 | 0.862 | 0.963† | **0.964**† |
| | RelFluency | 5.135 | 0.383 | 0.699 | 1.645 | 1.110† | **0.993**† |
| | Runtime (sec/edit) | 0.295 | **0.192** | 1.130 | 0.652 | 0.352 | 0.352 |
| FEVER | FlipRate | 0.868 | 0.997 | 0.979 | 0.941 | **0.999**† | **0.999**† |
| | CosSim | 0.900 | 0.844 | 0.883 | 0.894 | **0.944**† | 0.939† |
| | BERTScore-F1 | 0.761 | 0.840 | 0.856 | 0.865 | **0.949**† | 0.944† |
| | RelFluency | 3.508 | 0.623 | 0.706 | 0.756 | **0.972**† | 0.834† |
| | Runtime (sec/edit) | **0.251** | 0.255 | 1.428 | 0.401 | 0.364 | 0.364 |
| SciFact | FlipRate | 0.690 | 1.000 | 0.986 | 0.990 | **1.000** | **1.000** |
| | CosSim | 0.695 | 0.838 | 0.869 | 0.838 | 0.963 | **0.967**† |
| | BERTScore-F1 | 0.828 | 0.850 | 0.843 | 0.855 | 0.962† | **0.966**† |
| | RelFluency | 5.405 | 0.386 | 0.316 | 1.491 | 1.478 | **1.370**† |
| | Runtime (sec/edit) | 1.598 | **0.576** | 1.639 | 1.935 | 0.828 | 0.828 |

compute fluency and use roberta-base-uncased [20] to compute BERTScore-F1, as recommended by prior works [36, 66]. For fair comparison of latency, all of experiments are conducted on a single AWS instance p3.2xlarge with 61 GiB of memory and a single V100 GPU with 16 GiB VRAM. Our code implementation will be made public once this paper gets accepted.

**Evaluations.** We conduct evaluation with the suite of automatic evaluation metrics discussed in Section 4.2. In addition, we include human evaluation, details of which will be elaborated in Section 7.2. We leave qualitative studies in Appendix A.

## 7 RESULTS AND ANALYSIS

We discuss results for automatic evaluation metrics in Section 7.1 and human evaluation results in Section 7.2. We include more ablation studies in Section 7.3.

### 7.1 Main Results

**CFE2 can improve Max #Flip.** We can observe from Table 4 that on all datasets, CFE2 attains better FlipRate compared to baselines. For example, CFE2 can reach 0.999 FlipRate on FEVER and NQ, and 0.998 FlipRate on MS MARCO compared to the strongest baseline MaxFlip's 0.997, 0.996, 0.986. Interestingly, we find that after finetuning on the target corpus, the performance of **CFE2+** actually

downgrades by 0.001 on NQ. A potential reason is that finetuning makes the model pick up the specific writing patterns in the target collections and makes it hard to distinguish between hard negatives at the top ranks of SERP, leading to a slight degradation in FlipRate.

**CFE2 can improve closeness.** We study the CosSim and BERTScore-F1 metric from Table 4. We notice that on all datasets, CFE2 edits achieve significantly higher semantic similarity to the initial queries than the baselines. For example, on FEVER dataset, CFE2 reaches 0.944 CosSim and 0.949 BERTScore-F1 compared to the best baseline GenEx's 0.894, 0.865. We also discover that finetuning degrades similarity. For example, CFE2+ has a downgrade of 0.005 on both CosSim and BERTScore-F1 on FEVER dataset. We hypothesize the reason might be the similarity models, i.e. RoBERTa-base for BERTScore and CL-DRD are trained on a slightly different data distribution compared to FEVER.

Given the fact that CFE2 operates by iteratively Masking → Predicting, one can easily see that the generated edits will be of similar length as that of the initial query. Moreover, since we are replacing the masked tokens using algorithms that are trained on the corpus, the meaning and lexical/syntactic structure will also be similar to the initial queries. Therefore we do not include lexical similarity metrics such as BLEU [30], ROGUE [17] and Lexical Tree Similarity [65] for fair comparison with baselines. However, we do include lexical similarity in our human evaluation (Section 7.2).

**CFE2 can generate edits of similar fluency to the initial queries.** Recall our desiderata that the counterfactual query should be fluent (low perplexity) by itself and should have similar perplexity to the initial query (details in Equation (1)). From Table 4 we can observe that CFE2 can generate edits that are close to 1.0 in Fluency. CFE2 finetune reaches 0.972 fluency on FEVER dataset compared to best baseline GenEx's 0.756.

**CFE2+ consistently improves in terms of Fluency on all datasets.** This is expected as finetuning makes word prediction models predict more coherent tokens, leading to the low perplexity of the sequences. Between baselines, we notice that `MaxFlip` consistently outputs absolutely fluent counterfactuals (low perplexity). This is because it directly selects the lowest perplexity sentence from counterfactual document that can flip the pairwise relevance, leading to low perplexity results.

**CFE2 are faster in wall clock time, compared to other generation-based methods.** Ideally, explanations on SERP should have low computation complexity to pair with the low latency search system, and to also enable *interactivity* and *actionability*. We notice on all datasets, `Mask-only` and `MaxFlip` consistently achieve lower latency. This is expected as they do not rely on the editing/generation process and require significantly less computing. Compared to the generation-based baselines, i.e. `Summary` and `GenEx`, CFE2 achieves significantly faster inference time, which suggests the efficacy of the proposed editing-based approach.

## 7.2 Human Evaluaiton

We additionally include human evaluation to (1) validate the consistency between the proposed suite of automatic evaluation metrics (Section 4.2) and (2) examine the quality of counterfactual queries generated by CFE2.

**Human evaluation setup.** Annotators are recruited through MTurk; each annotator was given 50 $(q, d, d')$ triplets randomly sampled from SciFact dataset to simulate challenging web search queries. For each triples, annotators were asked multiple questions comparing the queries generated by the two methods i.e. CFE2+ and closest baseline method GenEx. They were asked to rate each of the queries compared to the initial queries on the following metrics: (a) **fluency** i.e. query being natural, grammatically correct and likely to be generated by a human [38], (b) **clarity**—how easy it is to understand the intent behind the query [37], (c) **closeness-semantic**, i.e. how semantically (i.e. in meaning) is the query close to the initial query, and (d) **closeness-syntactic**, i.e. how syntactically (i.e. in syntax) is the query close to the initial query. We show a sample screen in Figure 2. For each triplet, we collected at least three qualified responses, and the Fleiss' Kappa agreement rate for four metrics ranges from 0.08 (slight agreement) to 0.22 (fair agreement).

**Human evaluation results.** We report the human evaluation in Table 5. We can observe that among all four metrics, CFE2 edits significantly outperform edits from GenEx, and the improvement is statistically significant. These results are in alignment with automatic metrics from Table 4 and validate that the proposed suite of automatic metrics (Section 4.2) are indeed a good reflection of the generated counterfactual queries' quality.

**Respond for Query 1**

*How fluent is the query on its own?*
view rubric for Fluency

The query does not flow well. ○ ○ ○ ○ ○ The query is perfectly fluent.
                1  2  3  4  5

*How clear is the query to understand on its own?*
view rubric for Clarity

The query is illogical. ○ ○ ○ ○ ○ The query has no incorrect statements.
              1  2  3  4  5

*Do you think alternate query is semantically close to the original query?*
view rubric for Semantic Closeness

Strongly Disagree. ○ ○ ○ ○ ○ Strongly Agree.
              1  2  3  4  5

*Do you think alternate query is syntactically close to the original query?*
view rubric for Syntactic Closeness

Strongly Disagree. ○ ○ ○ ○ ○ Strongly Agree.
              1  2  3  4  5

*How weird is the query on its own? (Choose 4)*
view rubric for Fluency

The query is weird?. ○ ○ ○ ○ ○ The query is perfectly normal.
             1  2  3  4  5

**Figure 2: Screenshot of Annotation Task. The last question functions as an attention check.**

**Table 5: Results from human evaluation. p-value is calculated by Wilcoxon signed-rank test [59].**

| Metric | GenEx | CFE2 | p-value |
|---|---|---|---|
| Fluency | 3.76±0.97 | 3.83±0.93 | 1.25e-11 |
| Clarity | 3.59±0.92 | 3.88±0.95 | 1.51e-100 |
| Semantic Closeness | 3.12±1.12 | 3.64±0.88 | 2.50e-198 |
| Syntactic Closeness | 3.35±1.08 | 3.61±0.87 | 8.73e-61 |

## 7.3 Ablation Studies

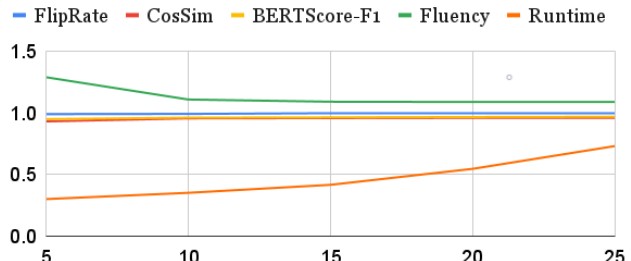

**Figure 3: Effect of beam size on FiQA dataset where x-axis denotes beam size. Runtime is measured by #seconds/edit. The line of CosSim overlaps the line of BERTScore-F1.**

**Effect of beam size.** In Figure 3, we study the sensitivity of the performance due to the change in beam size. We observe that as beam size increases from 5 to 10, the fluency quickly converges to 1. In addition, CosSim, BERTScore-F1 and FlipRate only achieve minor improvement. Yet, the runtime increases as beam size increases, since each word prediction takes $b \times b$ forward passes. Thus, we use $b = 10$ for all our experiments. The figure shows that different beam sizes do not affect end metrics significantly, showing the robustness of the framework to the hyperparameters.

**Effect of rank positions on CFE2.** In Table 6 we show a comparison of editing performance on different rank positions of the counterfactual documents. For all rank positions, we observe no significant difference in terms of FlipRate. In fact, CFE2 only fails to flip the pairwise relation in 1 out of 1148 instances. In terms of CosSim and BERTScore-F1, we observe a trend that similarity decreases along ranks, although the difference is minimal. This

**Table 6: Effect of rank positions on CFE2 's (w/o FT) editing performance on FiQA dataset, the rank positions of counterfactual documents are 2-5 respectively. We highlight the best numbers.**

| Counter. Doc. Rank | FlipRate | CosSim | BERTScore-F1 | RelFluency |
|---|---|---|---|---|
| 2 | 1.000 | **0.964** | **0.969** | **1.062** |
| 3 | 1.000 | 0.960 | 0.966 | 1.073 |
| 4 | 1.000 | 0.958 | 0.964 | 1.145 |
| 5 | 0.999 | 0.956 | 0.963 | 1.086 |

pattern is anticipated since the original ranking algorithm CL-DRD relies on Maximum Inner Product Search (MIPS) to construct the ranklists, thus the lower ranked documents are less similar to the query, leading to less similar counterfactual queries. We also observe that Fluency slightly degrades along ranks. One possible interpretation is that the lower-ranked documents are less similar to the original query, leading to less similar counterfactual queries in Fluency (higher perplexity compared to the original query).

**Table 7: Performance Comparison between different word prediction models. Beam size is set to 10 for both models.**

| Dataset | Metric | DistilBERT w/o FT | BERT w/o FT |
|---|---|---|---|
| FiQA | FlipRate | 0.999 | 0.999 |
| | CosSim | 0.957 | 9.959 |
| | BERTScore-F1 | 0.963 | 0.960 |
| | RelFluency | 1.110 | 1.055 |
| | Runtime (sec/edit) | 0.352 | 0.503 |

**Effect of different word prediction models.** One concern that readers may have is that our editor model and search model both use DistilBERT as the backbone, and this may challenge the model-agnostic claim of the proposed methodology. In addition, we also would like to know the effect of different word prediction models on the editing performance. We also experimented with off-the-shelf BERTForMaskedLM and report its performance in Table 7. We can observe that BERT only achieves minor improvement compared to DistilBERT in terms of CosSim, BERTScore-F1 and Fluency, and have the same performance in terms of FlipRate on FiQA dataset; the same pattern holds for the rest three datasets as well. In contrast, we observe CFE2 based on BERT is significantly slower than DistilBERT, e.g. it takes 42.8% more time per edit on FiQA dataset, as bert-base-uncased has 110M parameters compared to distilbert-base-uncased's 66M. Therefore, we conclude that CFE2 with DistilBERT better meets our low latency desiderata.

## 8 CONCLUSION AND FUTURE WORK

In this work we study the effectiveness of counterfactual explanations for search engine result explanation task. First, we discuss the possible formulations of counterfactual explanation in the context of web search. Next, we discuss desiderata for counterfactual explanations in search system and operationalize them with appropriate metrics. Ultimately, we propose **CFE2** , a framework to automatically generate counterfactual queries by editing the initial queries. The proposed method achieves significant improvement compared to baselines on both proposed evaluation metrics and human evaluation. For future work, we hope to verify the effectiveness of the **CFE2** for the end use case of query suggestion/reformulation in online experiments.

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

# A QUALITATIVE STUDIES

Here we showcase a few examples of the success and failure of the proposed CFE2 . All these examples are from MS MARCO dataset dev subset.

> **Initial query**: what is process control equipment
> **Top-1 document**: what is process control? process control is an algorithm that is used in the during the manufacturing process in the industries for the active changing process based on the output of process monitoring.
> **Counterfactual document**: process equipment is equipment used in chemical and materials processing, in facilities like refineries, chemical plants, and wastewater treatment plants. this equipment is usually designed with a specific process or family of processes in mind and can be customized for a particular facility in some cases.
> **Counterfactual query**: what is process ~~control [PAD]~~ equipment

In the above example, we can see CFE2 captures the topic difference between top ranked document and 2nd ranked document, and it provides a reasonable explanation for the pairwise relevance relation, i.e. to remove the keyword **control**.

> **Initial query**: cost of attendance eastern illinois university
> **Top-1 document**: eastern illinois university has roughly 8,000 students. admission is selective. tuition is approximately $8,550 per year for residents of illinois and other bordering states, while it is $10,680 for non-residents. additional fees amount to $2,762.32. the university estimates its average cost-of-attendance to be approximately $24,640 per academic year. tuition is expected to increase in the 2016 - 2017 academic year.
> **Counterfactual document**: the cost of attending northern illinois university for in-state students without financial aid is $14,295. the cost for out-of-state students without financial aid is $23,761.
> **Counterfactual query**: cost of attendance ~~eastern~~ **northern** illinois university

In the above example, we can see the 3rd ranked document is about northern illinois university, and CFE2 successfully captures the difference and suggests a reasonable and informative counterfactual query.

> **Initial query**: how long is a typical car loan?
> **Top-1 document**: in general, car loans are structured to offer 12 - month increments and last somewhere between two and eight years. that means you'll find available loans of 24 months, 36 months, 48 months, 60 months, 72 months and 84 months. the average new car loan is around 65 months, or more than five - and - a - half years, while the average used car loan is shorter. long - term drawbacks. when you're signing the paperwork at the dealer, you'll be tempted to go for a longer term
> **Counterfactual document**: if you're in the market for a new car, the length of the average auto loan may surprise you. loans for many years were typically around five years, or 60 months. buyers now seek varying loan lengths and terms, depending on the vehicle and the state of the economy at the time of purchase.
> **Counterfactual query**: how long is a typical ~~car~~ **automobile** loan?

In the above example we can see although CFE2 captures the keyword **auto** in the counterfactual document, it fails to predict an informingly different query. This is partially due to the similarity between the top-1 ranked document and the lower ranked counterfactual document. This suggests that when the counterfactual document also addresses the information need of the initial query, CFE2 may fail to produce a sufficiently different counterfactual query as explanation.

