# OpenReview forum: "CFE2: Counterfactual Editing for Search Result Explanation"
_ACM.org/SIGIR/ICTIR/2024/Conference — ICTIR 2024_

### Official Review · Reviewer_zA7J · 2024-05-17

**Rating:** 1
**Confidence:** 4

**Objective Part Of Review:**

Motivated by counterfactual thinking, this submissions uses alternate queries as an explanation of why two documents are ordered the way they are in a ranked list -- or, more accurately, why one is ranked above the other and what would need to change to flip their order. The authors develop an algorithm to create these counterfactual queries and evaluations (including asking people) to measure whether they're successful.

I really like the motivation of the paper and the ideas underlying it. However, the biggest challenge to me was figuring out how this would be used. What would it look like as an explanation in a search result page? In that context, is it meaningful or understandable? The evaluation shows that something good is happening with a _pair_ of documents, but what if there are 10 documents: will all 90 pairs be explained? Or just select ones? I still really like the paper, but the lack of explanation for the explanation left me somewhat less enthusiastic.

 Here are some points that I found a little confusing or was unsure about:

* The problem definition should probably somehow say that $rel(q',d)$ is still reasonable. That is, not just that $q'$ is has a strong connection to $d'$ but that it also maintains some degree of connection with $d$.
* In Algorithm 1, what is $n$? It is not a parameter listed. Later you refer to the $n$ important tokens; is that it?

**Subjective Part Of Review:**

Here are some small points that I hope the authors will consider, though none of them is a deal-killer:

* In the introduction, the RQ's are not actual questions. It's a little confusing.
* At the start of section 2, what do you mean by "the final version"?!?
* When defining the problem, there should be no comma after problem: "We formally define our problem as follows:"
* In Algorithm 1, $n$ appears to be a count. But in equation two it is listed as $|n|$ which suggests that it is a set or a number that might be negative.
* In 7.2's header, Evaluation is misspelled as Evaluaiton

---

### Official Review · Reviewer_7JXf · 2024-05-20

**Rating:** -2
**Confidence:** 4

**Objective Part Of Review:**

This submission describes and evaluates a technique to explain search results: specifically by generating an alternative query, under which some other document would be higher-ranked. Explanations of search results are perhaps not important for most searchers/searches, but are undeniably important in some cases and still not well-supported, so this could be useful work.

*Presentation*: the paper is generally well-written and easy to follow (but see some specific comments below). It's laid out well and the presentation of the key algorithm and steps makes sense. There is a good level of detail.

*Related work*: there are some useful pointers into the literature.

*Scope*: the experiments here are motivated by work from cognitive science, and clearly applied to core and current IR problems.

*Soundness*: this is an interesting general topic, but unfortunately the contribution of this paper isn't immediately clear. Some clarifications would help: for example, what use case are we supporting here? There are a few different audiences for explanations (searchers, regulators, researchers, ...) and even assuming it's the searcher we're interested in, there are cases when explanations would be more or less useful and a few goals they might support (building confidence in the results, helping write better queries, helping explore a set of documents, improving a mental model, ...). This isn't really articulated anywhere, which makes it hard to know whether the proposal works well or where it's interestingly good or bad. This also means the desiderata in s4.1 are debatable: for example, none of them say anything about a searcher learning anything and rather they say something about the form of the algorithm.
Focussing on a counterfactual _query_, not _document_, makes sense and is a clever idea if the goal is to support reformulation. If the goal is something else, it's less clear from the present text.

Examples would certainly help. There is one example interaction, in Fig 1(a), but it's not described in these terms and it takes some detective work to find (it's also only partial as no example document is included). Importantly that example shows (some of) a final pair {q', d'}, but not how that might be exposed as an explanation for the searcher - or regulator or whoever. From that example, it seems we can say "if you'd asked about a different battle, you'd have got a different answer" - but as a searcher I already know that, or at least I expect that! Modifying queries in the way proposed does seem to tie us to this sort of topical explanation, which rules out non-topical factors such as date, authority, link structure, popularity, or any of the countless other things used in a modern search engine.

The evaluation is straightforward, but some open questions make the results hard to interpret. The note on desiderata above is relevant here: yes, the algorithm is good at (e.g.) BERTscore-F1, so $q'$ is similar to $q$, but is that what we want from an explanation at the end of the day? It would be nice to consider evaluation by humans (as here), but on metrics closer to final human use. (At line 868, the results don't reflect the explanation's quality: they just mean human and automated metrics agree. We can't say whether both metrics are valid, or both are invalid.)

The choice of baselines here also obscures the contribution. These aren't other good ways to explain a result; rather they're other ways to generate the same intermediate steps. Again that makes it hard to know how this compares with alternatives.

Some smaller notes:

* line 347: is this one "counterfactual document" or just one "document"? It's a real document from further down the ranking.
* s4.1: there are no restrictions here on the nature of $d'$, but presumably we want it to be as close to $d$ as possible and vary only on some salient aspect. This is constrained by the retrieval, i.e. there's not likely to be anything really different in the top results, but it might be worthwhile spelling it out explicitly (if that's indeed correct).
* is GenEx a fair baseline if it's not trained per the original work?

**Subjective Part Of Review:**

This is an important problem, and the work here could be useful in many places. It would be much stronger though if there were some explicit, well-argued connection between what we want an explanation to do and the properties being measured here.

Since there are humans providing data, it would be nice to see a short note about IRB approval (or whatever process applies locally).

---

### Official Review · Reviewer_AkPE · 2024-05-22

**Rating:** 1
**Confidence:** 4

**Objective Part Of Review:**

Summary: This work proposes the task of creating counterfactual explanations for search results. In this context, a counterfactual explanation indicates changes to the query that would flip a given pair of documents. In practice, the task is to generate a new query that changes the order of two documents while fulfilling several reasonable conditions intended to ensure the explanation is useful (e.g., ensuring the new query is as close as possible to the original query). The work proposes this type of explanation in the context of ranking, outlines desiderata that a good explanation (new query) should have, and proposes an approach for generating a new query that fulfills these desiderata. The approach is evaluated using automatic evaluation metrics proposed in the work and using human judgments from crowdworkers. In the automatic evaluation, the proposed approaches perform well in terms of flip rate, closeness metrics, and relative fluency compared to the original query. In the human evaluation, the proposed approach is generally preferred over the strongest baseline in terms of average metrics, though the standard deviations reported are relatively large.

This work clearly describes the problem considered, the proposed method for generating counterfactual queries to serve as explanations, and the evaluation process. There are several aspects of the evaluation that weaken the results presented:
- The MaxFlip evaluation metric is directly optimized by the proposed approach (line ~617) on the same data, so including it as an evaluation metric is unfair. The baselines do not seem to have this advantage. The results show that the proposed approach almost always satisfies this metric (e.g., 0.998 on MS MARCO).
- In general, the evaluation metrics are very high across both variants of the proposed method and the strongest baselines (e.g., 0.910-0.933 for CFE2+ compared to 0.902-0.878 for GenEx on MS MARCO). This suggests they may not be the best way to evaluate this task, unless the task is already solved.
- Inter-annotator agreement is low for the human evaluation (Kappa between 0.08 and 0.22), which is also reflected in the standard deviations reported in the table (e.g., 3.76 +- 0.97 compared to 3.83 +- 0.93 for GenEx and CFE2 fluency).
- The examples in the Appendix highlight some issues, which may be related to how closeness is operationalized. Changing "car" to "automobile" appears to result in a similar query, whereas the example changing "eastern illinois university" to "northern illinois university" substantially changes the meaning of the query, even though only a single term is modified and "eastern" is semantically similar to "northern" in some sense.

**Subjective Part Of Review:**

This is an interesting work that outlines a nice new direction for explaining search results. It convincingly motivates the counterfactual approach taken. The problem definition and proposed direction are the most compelling parts of the paper to this reviewer. While the proposed method performs well in the evaluation, there are several issues that limit how exciting this is (see objective part). The paper itself is easy to follow and I would expect it to be of interest to the community despite its limitations, though it would be much better to acknowledge the evaluation limitations in the paper.

---

### Official Review · Reviewer_26kB · 2024-05-22

**Rating:** 2
**Confidence:** 4

**Objective Part Of Review:**

The paper investigates the impact of counterfactual explanations for web search results and proposes to use counterfactual queries as explanations.

The discussion of related work is comprehensive and the work is well embedded in the body of literature.

A list of desiderata is presented along with a corresponding set of automatic evaluation measures. Based on the desiderata, an approach for generating counterfactual queries as explanations is presented. The proposed approach is compared against a set of appropriate baselines.

The automatic results are complemented with human evaluation and an ablation study.

Overall, the paper is comprehensive and very nicely done, it ticks all the boxes.

The only part that leaves me with questions is the low agreement between human judges. It would be nice to have some more details on that.

**Subjective Part Of Review:**

Why this specific list of desiderata (in Section 4.1) and not more/less/others? Some more motivation would be nice.

Human evaluation focuses on the validity of the automatic measures and the quality of queries. I would have expected also to ask users about their perceived usefulness of such explanations.

Nit: typo in the heading of Section 7.2

---

### Meta-Review · Area_Chair_Ze2T · 2024-05-24

**Recommendation:** Accept (Oral)
**Confidence:** 4

**Metareview:**

This is a meta-review. This paper proposes a method for search result explanation based on counterfactual query editing. The reviews are mixed, but three out of the four reviewers would like to see this paper accepted for ICTIR. The main strengths of the paper are the motivation and idea, the good problem definition, and the broad evaluation including a user study. Weaknesses are the low agreement between human judges, the evaluation metrics used (e.g. the MaxFlip metric is directly related to the proposed method), and the utility of the work in a practical search setting. In addition, examples would help understand the intuition of the method and again, the utility of it in practice. Reviewer 7JXf provides a relevant discussion of this issue that I recommend the authors to incorporate in their revision. After discussion with the four reviewers, my standpoint is that this paper has value to the community because: (a) explainability is underexplored in search engines compared to recommender systems; (b) the limitations in terms of practical utility that is pointed out by the reviewers is not a major concern in this stage of the work, especially not for ICTIR; (c) the idea of counterfactual generation is interesting and deserves further discussion and exploration during the conference and in follow-up work.